# Kramer Score, an Evidence of Its Use in Accordance with Indonesian Hyperbilirubinemia Published Guideline

**DOI:** 10.3390/ijerph18116173

**Published:** 2021-06-07

**Authors:** Mahendra Tri Arif Sampurna, Muhammad Pradhika Mapindra, Muhammad Pradhiki Mahindra, Kinanti Ayu Ratnasari, Siti Annisa Dewi Rani, Kartika Darma Handayani, Dina Angelika, Agus Harianto, Martono Tri Utomo, Risa Etika, Pieter J. J. Sauer

**Affiliations:** 1Neonatology Division, Department of Pediatrics, Dr. Soetomo Academic Teaching Hospital, Faculty of Medicine, Universitas Airlangga, Surabaya 60285, Indonesia; kartika09rama@gmail.com (K.D.H.); dina.angelika@yahoo.co.id (D.A.); neonatologi.soetomo@gmail.com (A.H.); mrmartono73@gmail.com (M.T.U.); risa_etika@yahoo.com (R.E.); 2Neonatology Division, Department of Pediatrics, Airlangga University Teaching Hospital, Faculty of Medicine, Universitas Airlangga, Surabaya 60115, Indonesia; 3Neonatal Research Group Surabaya, Faculty of Medicine, Universitas Airlangga, Surabaya 60285, Indonesia; mpradhika57@gmail.com (M.P.M.); pradhikim@gmail.com (M.P.M.); kinanyuu@gmail.com (K.A.R.); sannisadr@gmail.com (S.A.D.R.); 4Department of Pediatrics, Beatrix Children’s Hospital, University Medical Center Groningen, University of Groningen, 9713GZ Groningen, The Netherlands; saupie46@gmail.com

**Keywords:** hyperbilirubinemia, Kramer score, Indonesia guideline, phototherapy, reliability

## Abstract

**Background:** In some hospitals in low/middle-income countries, methods to determine the bilirubin level in newborn infants are unavailable and based on a clinical evaluation, namely a clinical score designed by Kramer. In this study, we evaluated if this score can be used to identify those infants that need phototherapy. **Method:** Infants admitted between November 2018 and June 2019 to three hospitals in Surabaya, Indonesia were included. The jaundice intensity was scored using the Kramer score. Blood was sampled for total serum bilirubin (TSB) measurement. The infants were categorized into Treatment Needed (TN) group when treatment with phototherapy was indicated and the No Treatment Needed (NTN) group when phototherapy was not indicated, based on the Indonesian Guideline for hyperbilirubinemia. **Result:** A total of 280 infants with a mean birth weight of 2744.6 ± 685.8 g and a gestational age of 37.3 ± 2.3 weeks were included. Twenty-seven of 113 (24%) infants with Kramer score 2 needed phototherapy, compared with 41 of 90 (46%) infants with score 3 and 20 of 28 (71%) of infants with score 4. The percentage of infants that needed phototherapy was higher with decreasing gestational age. **Conclusion:** The Kramer score is an invalid method to distinguish between those infants needing phototherapy and those infants where this treatment is not indicated.

## 1. Introduction

The majority of newborn infants develop jaundice in the first week after birth due to an increased bilirubin level. In most infants, the bilirubin levels decrease spontaneously and are harmless. [1] In a minority of cases, the bilirubin increases to potentially dangerous levels. High bilirubin levels may cause acute or chronic brain damage. [2] Brain damage due to high bilirubin levels is presently rare in High-Income Countries (HIC). Brain damage can be prevented by early recognition of infants at risk and subsequent treatment with phototherapy. [3] The decision to start phototherapy is in HIC based on the bilirubin level, either measured in blood or transcutaneously. The incidence of bilirubin-induced brain damage is still high in Low-Middle Income Countries (LMIC). It is often not possible in remote areas of LMIC to measure the bilirubin level in newborn infants because laboratories, when present, may lack the apparatus needed [4]. Transcutaneous systems are not available due to the high costs and problems with maintenance [5]. Health care workers might therefore base their decision to administer phototherapy on a visual inspection of the infant. The visual inspection is mostly based on the Kramer score [6], although this score was not designed to identify those infants who need phototherapy. In this study, we evaluate if the Kramer score can distinguish between those infants who need phototherapy and those where it is not indicated.

## 2. Materials and Methods

This prospective observational study was conducted in three hospitals, Dr. Soetomo Hospital Surabaya, Malang Hospital, Malang, and the Women’s Children’s Hospital in Surabaya, East Java, Indonesia. Participants were recruited between November 2018 and June 2019. All infants were from the same ethnic background, Malay-Mongoloid with Fitzpatrick’s scale of both parents 3–4. Inclusion criteria of this study were neonatal jaundice assessed as a Kramer’s scale of at least 1, gestational age (GA) of ≥32 weeks or birth weight of ≥1500 g, and postnatal age <15 days. Exclusion criteria were: congenital abnormalities, cardio or respiratory insufficiency, and phototherapy in the past 24 h. The study was approved by the Institutional Review Board (or Ethics Committee) of the Clinical Research Unit at Dr. Soetomo General Hospital (no. 0526/KEPK/VIII/2018) and Universitas Airlangga Hospital (192/KEH/2018). Both parents filled in the consent form. 

In this study, we categorized jaundice according to the Kramer score. The Kramer score defines the intensity of jaundice based on the dermal advancement from head to hands and feet [6]. Kramer score 1 represents jaundice of head and neck, 2, trunk to the umbilicus, 3, groin including upper thighs, 4, knees and elbows to ankle and wrists. 5, feet and hands including palm and soles. [6] The scoring of the Kramer scale must be done in natural light by blanching the skin. Infants in Dr. Soetomo hospital were scored by residents in pediatrics who had followed a course on how to use the Kramer score. In the two other hospitals, the scoring was done by nurses who were trained to use the Kramer score. Directly after the clinical scoring of the infant blood was taken for the measurement of Total Serum Bilirubin (TSB). The bilirubin level at all sites was measured with Dimension EXL 2000. The TSB was plotted on the graph of the recently introduced Indonesian guideline for the detection and treatment of hyperbilirubinemia [7,8]. This graph shows the level of TSB where providing phototherapy is indicated. The graph takes into account the birthweight and the postnatal age of the infant. When the TSB was above the line for phototherapy, the infant was included in the group Treatment Needed (TN), when the TSB was below that line the infant was included in the group No Treatment Needed (NTN). For plotting the TSB on the line for phototherapy, we used the mobile application integrated with the Indonesian hyperbilirubinemia guideline called the BiliNorm (http://bilinorm.babyhealthsby.org, accessed on 29 March 2021). Because information on risk factors for developing severe hyperbilirubinemia or the presence of diseases that makes infants more vulnerable for brain damage, was not available, all infants were categorized as having unknown risk factors. This means that, according to the Indonesian guideline, all infants were presumed to have risk factors. [8] The decision to categorize the infants as having a risk factor was also made because it is impossible to identify in remote areas in infants the presence of risk factors. 

### Data Analysis

Data were analyzed using IBM Statistic SPSS Version 21.0 (IBM., Corp., Armonk, NY, USA) Descriptive analysis was calculated to obtain mean (M)*,* standard deviation (SD), and 95% confidence interval (CI) with the range. We divided the infants into three groups according to gestational age, <34 weeks, 34–37 weeks, and >37 weeks. 

## 3. Results

Three hundred twenty-nine newborn infants were included in this study, 119 infants were included in Dr. Soetomo hospital, 64 in the Malang hospital, and 97 newborns were enrolled from Women’s Children’s Hospital. Forty-nine infants were excluded, 21 infants showed respiratory or circulatory insufficiency, 12 infants had a birth weight <1500 g, 2 infants were diagnosed with multiple congenital anomalies, and 14 infants were excluded because no consent was given by the parents. Therefore, 280 infants met the eligibility criteria for this study, with a mean birth weight of 2744.6 ± 685.8 g. The characteristics of the newborns are presented in Table 1. The majority of the infants were born at term (56.1%). Forty-two percent of newborns were assessed by pediatric residents, 58% were scored by nurses. 

The TSB level of all infants and the Kramer score of these infants is shown in Figure 1. There is a wide range of bilirubin levels at each Kramer score, as well as a wide overlap of bilirubin levels between the Kramer scores. Figure 2 shows, for each of the Kramer scores, the number of infants, who, according to the recent Indonesian guideline, needed phototherapy and those that did not need phototherapy. Only 3 out of 48 infants with Kramer score 1 needed phototherapy, while 27 out of 113 infants at score 2 needed phototherapy. Almost half of the infants at score 3 needed phototherapy and 71% of infants at score 4. There was only one infant with a score of 5, this infant needed therapy. In Figure 3, we show the data according to gestational age. In infants with a gestational age (GA) of <34 weeks, 1 out of 5 infants (20%) with Kramer score 1, 2 of 6 infants (33%) with score 2, 5 of 6 infants (83%) with score 3, 4 of 5 infants (80%) with score 4, and one infant with score 5 (100%), required phototherapy. Although the numbers are relatively small, it shows that almost all infants <34 weeks needed phototherapy at Kramer score 3, 4, and 5, while one-third of infants at score 2 needed therapy. The percentage of infants in the group 34–37 weeks that needed phototherapy with the Kramer score 1, 2, 3, and 4 were respectively 13, 23, 54, and 44%. In infants born >37 weeks, 24% of infants with a score of 2 needed phototherapies, 35% with a score of 3, and 86% with a score of 4. It was noteworthy that none of the infants with GA > 37 weeks and Kramer 1 required therapy. When expressed differently, 76% of term infants with a score of 2 did not require phototherapy, 65% with a score of 3, and 14% with a score of 4. 

## 4. Discussion

According to our results, the Kramer score cannot differentiate between those infants where phototherapy is indicated vs those infants where PT is not indicated. Compared to the original paper of Kramer [6] our findings showed a much wider range of bilirubin serum levels at each Kramer score. In 1969 Kramer described that the advancement of jaundice in the dermal zones of newborn infants is related to the serum bilirubin level. Jaundice was scored from no jaundice, score 1 to the jaundice of palms and feet, score 5. According to these results, Kramer concluded that a score of 2 indicated that the bilirubin level was below 12 mg/dL and a score of 3 below 15 mg/dL. There was no difference between the infants born at term and infants with a low birth weight regarding the relation between bilirubin level and score. There was however a wide range in serum bilirubin levels at each score. Kramer concluded that simple inspection of the skin may provide useful information in the management of infants with hyperbilirubinemia. Kramer also concluded that “Visual assessment of jaundice cannot replace the importance of laboratory investigation of serum bilirubin” [6]. In Kramer’s study, no distinction was made between preterm infants and growth-retarded infants, neither was the question raised or answered if the score could be used to identify those infants who might need treatment for the hyperbilirubinemia. Keren et al. in 2009 compared the Kramer score with the transcutaneous measured bilirubin in 522 infants [9]. They observed that the incidence of hyperbilirubinemia increased with an increasing Kramer score, but the Kramer score was not a good predictor for hyperbilirubinemia, defined as the level needed phototherapy. They found that 6% of infants with Kramer score 2–3 developed hyperbilirubinemia and needed phototherapy, and 10% of infants with Kramer score 4–5. The majority of infants in this study had rather low bilirubin levels, only 4% of included infants developed hyperbilirubinemia [9]. We found a higher percentage of infants who needed phototherapy, in term infants, 24% of infants with Kramer score 2 and 35% of infants with Kramer 3. If this difference is due to a different scoring, a difference in skin color or lower levels for providing phototherapy is not clear. Tikmani and Warraich studied the incidence of hyperbilirubinemia in a region of Karachi. Infants were visited at home at different time points from 48–72 h to 6–9 weeks. Twenty-eight percent of the cohort of 1690 infants were referred to a local primary care clinic because of jaundice. It was estimated that 6% of all infants in that area had a bilirubin level >15 mg/dL. They found that only one infant out of 42 infants with a Kramer score of 2 had a bilirubin level >15 mg/dL, compared to 14 out of 54 with a score of 3 and 8 out of 13 with a score of 4. Serum bilirubin was measured in only 30% of infants. Many parents refused blood taking. [10].

Okwundu and Saini recently reviewed the methods to measure and estimate levels of bilirubin in newborn infants. They concluded that visual inspection, also when using the Kramer score, is the least reliable method. One of the reasons might be a low inter-observer agreement. Although they stated that the Kramer score of 1–2 had a sufficient negative predicted value, they concluded that visual inspection cannot be used solely for the assessment and management of neonatal jaundice and hyperbilirubinemia. [11].

We found that 24% of infants with a Kramer score of 2 needed phototherapy. In term infants, this was 21% and 33% in preterm infants. There might be two reasons why the percentage in our study is higher than reported in other studies. First, our children have a darker skin compared to Caucasian infants. Recent studies indicate that the transcutaneous bilirubin meter slightly overestimates the serum bilirubin in infants with a darker skin compared to Caucasian infants [12,13] We do not know if the darker skin color of infants in Indonesia might have caused an underestimation of jaundice, so a lower score at higher bilirubin levels. Secondly, low levels to start phototherapy were chosen, because we assumed that all infants might have risk factors for the development of increased bilirubin levels or might be more sensitive for bilirubin-induced encephalopathy. In remote areas, and also in many hospitals in Indonesia, the methods to distinguish between infants with or without a risk factor are not available. 

The percentage of infants needing phototherapy was higher for infants <34 weeks compared to term infants. These results are in agreement with the study of Szabo et al. [14] who found bilirubin levels in preterm infants up to 265 µmol/l while showing a Kramer score of 2 and 3. There was a wide overlap in bilirubin levels between infants with Kramer score 2, 3, 4, and 5. We conclude from our study and the study of Szabo that the Kramer score cannot be used to identify those preterm infants where phototherapy is indicated. 

The Kramer score is still used and advised by the World Health Organization (WHO) and the Indonesian Guideline on hyperbilirubinemia for remote areas where measuring either serum or transcutaneous bilirubin is not possible. We suggest that newborn infants under these conditions are screened daily by a trained health care worker. When the Kramer score increases from 2 to 3, phototherapy should be given. When the score is 4 or higher the infant must be transferred to a center where bilirubin levels can be measured. Daily visits must be continued until a safe score, 1, is obtained. 

There are different methods to prevent the bilirubin level to increase to dangerous levels. Phototherapy is the most widely used method. It is safe and easy to apply. Other methods are exchange transfusion and intravenous Immunoglobulin.

Phototherapy causes several biomolecular reactions, including photooxidation and photoisomerization. Phototherapy enables bilirubin to undergo the photooxidation process to become quite small and polar to be excreted via the urinary tract. The photoisomerization will result in the formation of lumirubin, a constitutional isomer of bilirubin, which is more readily excretable in bile [15]. Exchange transfusion, particularly rewarding for blood group incompatibility, is a substitution of recipients’ erythrocyte mass and plasma by blood from donors [16]. Immunoglobulins seem to be only useful in the case of severe alloimmune hemolytic disease [17]. 

A limitation of this study is that the scoring of the Kramer score was done by different examiners. They all were trained in applying the Kramer score. The study from Szabo et al. showed a good correlation between the scores of nurses and one researcher [14]. Moreover, there was no difference in the percentage of the different scores between hospitals. Finally, in clinical practice, the scoring also is done by different health care workers. 

## 5. Conclusions

In conclusion, we found that the Kramer score cannot be used in both preterm and term Indonesian infants to differentiate between newborn infants that need phototherapy or not. In all infants with a Kramer score of 2 and higher measuring the bilirubin level, either in serum or transcutaneously, is indicated. 

## Figures and Tables

**Figure 1 ijerph-18-06173-f001:**
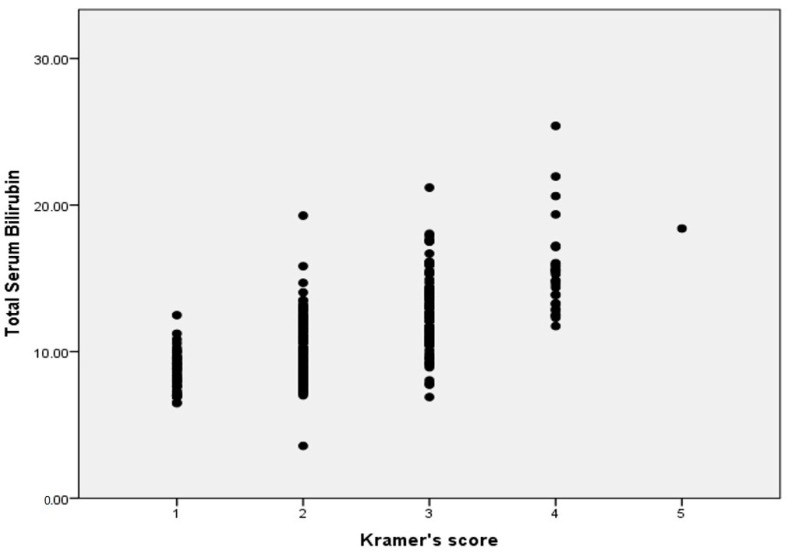
Serum Bilirubin Level in relation to the Kramer score for all infants. Each dot represents one infant.

**Figure 2 ijerph-18-06173-f002:**
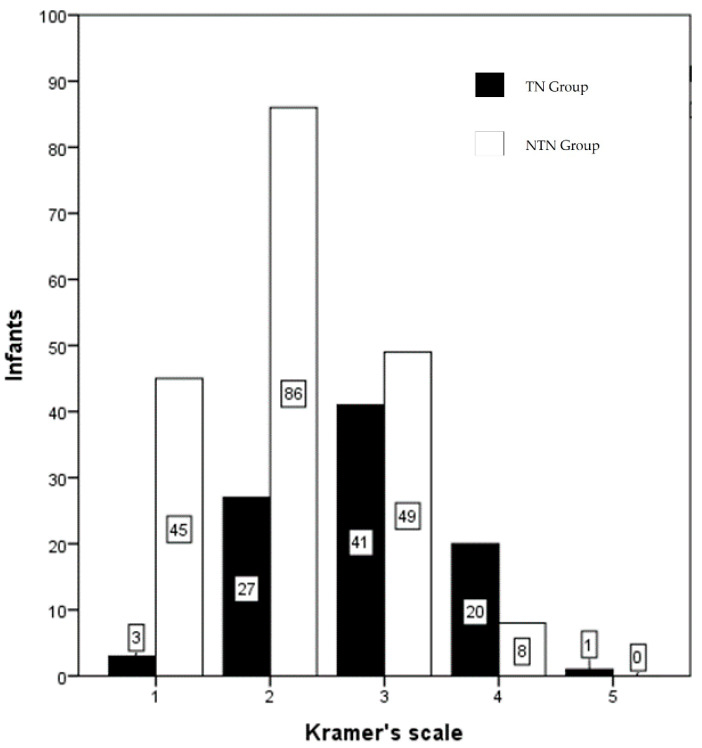
Number of infants needing phototherapy for each Kramer score. Black Bars infants who needed therapy (TN group); white bars infants where phototherapy was not indicated (NTN group).

**Figure 3 ijerph-18-06173-f003:**
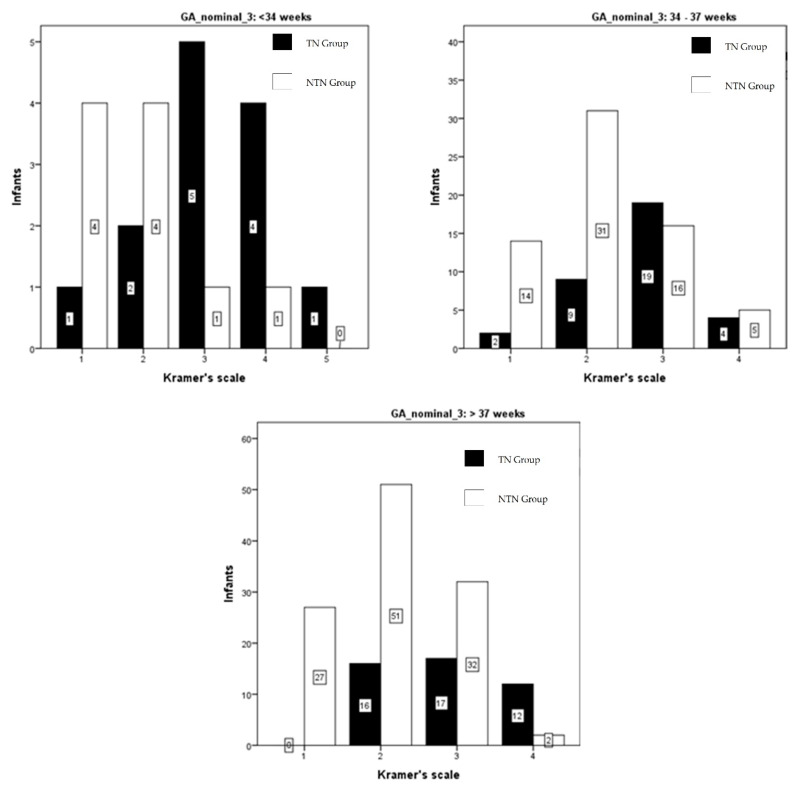
The number of infants needing phototherapy for each Kramer score in relation to gestational age. Black Bars infants who needed therapy (TN group); white bars infants where phototherapy was not indicated (NTN group).

**Table 1 ijerph-18-06173-t001:** Clinical characteristics of newborns.

Characteristics (*N* = 280)	Value
Gestational age (weeks)	37.3 ± 2.3 (32–42)
<34 weeks	23 (8.2)
34–37 weeks	100 (35.7)
>37 weeks	157 (56.1)
Birth weight (gram)	2744.6 ± 685.8 (1500–4700)
1500–1999 g	53 (18.9)
2000–2499 g	44 (15.7)
2500–2999 g	68 (24.3)
3000–3499 g	82 (29.3)
≥3500 g	33 (11.8)
Age (days)	3.9 ± 2.6 (1–14.8)
Age (hours)	93 ± 62.2 (24–355)
Total Serum Bilirubin (mg/dL)	11.31 ± 3.1 (3.57–25.4)

Data are presented as mean ± SD [range] or *N*(%).

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
