# Peer review of "Kramer Score, an Evidence of Its Use in Accordance with Indonesian Hyperbilirubinemia Published Guideline"

_ijerph, 2021, doi:10.3390/ijerph18116173_

Round 1

Reviewer 1 Report

Major comment

In this study the authors aim at evaluating if the Kramer score can distinguish between those infants that need phototherapy and those where it is not indicated. It is an interested study but the conclusion is not supported by the results. I suggest the authors should consult the statisticians and rearrange the results.

Authors report that Pearson Chi-Square test was used to determine the comparison between Kramer’s scale and TSB levels but the results are not presented.

Minor comments

Line 34: insert the abbreviation for Low-Middle Income Countries (LMIC)

Line 37: “their” instead of “its”

Line 39: insert reference for the Kramer score

Line 65: “all infants were categorized” instead of “were all infants categorized”

Line 78: “The TSB in each category of the Kramer score is shown in histograms” must be transferred to result section and reported as figure 1 in the text.

Lines 84, 85: please use Arabic numerals instead of “twenty-one” and “two”.

Line 88: spelling “The majority of newborn”

Lines 88-89: the mean birthweight refers to all the study newborns, not only to those born at term. Please rephrase the sentence.

Table 1: use footnotes for explanation of the data (n%?) and symbols.

Lines 100, 109: “who” instead “that”

Please use abbreviations when using a word for the first time in the text.

Line 138: level of bilirubin.

Line 151: blood taking? Better “blood collection”

Please check spelling and dictionary all over the text.

Author Response

Response to Reviewer 1 Comments

Point 1: In this study the authors aim at evaluating if the Kramer score can distinguish between those infants that need phototherapy and those where it is not indicated. It is an interested study but the conclusion is not supported by the results. I suggest the authors should consult the statisticians and rearrange the results.

Response 1: We thank the reviewer for her/his remarks. It helps us to improve our manuscript.

As suggested by the reviewer, we consulted a statistician. She agreed with the design of the study as well as with results as written in the paper. We apologize we might have not been clear enough in the description of the study and the results.

Our study is a descriptive study. For each infant the Kramer score was determined. The serum bilirubin was measured at the same moment. We then plotted the result of the serum bilirubin on the graph of the most recent Indonesian guideline for the management of hyperbilirubinemia. This graph indicates if the infant needed phototherapy or not. For the analysis all infants were grouped according to their Kramer score. We only used descriptive statistics, so no statistical calculation was performed. 

Point 2: Authors report that Pearson Chi-Square test was used to determine the comparison between Kramer’s scale and TSB levels but the results are not presented.

Response 2: We apologize for this mistake. We omitted the results of this comparison as it was not appropriate, but forgot to delete the description of this test from the paper.

Point 3: Line 34: insert the abbreviation for Low-Middle Income Countries (LMIC)

Response 3: Thank you for your suggestion. We have adjusted according to the suggestion

Point 4: Line 37: “their” instead of “its”

Response 4: Thank you for your remark and adjusted it.

Point 5: Line 39: insert reference for the Kramer score

Response 5: Thank you  for your suggestion. We inserted the reference

Point 6: Line 65: “all infants were categorized” instead of “were all infants categorized”

Response 6: Thank you  for your remark, we adjusted according to the suggestion.

Point 7: Line 78: “The TSB in each category of the Kramer score is shown in histograms” must be transferred to result section and reported as figure 1 in the text.

Response 7: Thank you for your suggestion. We changed the text accordingly.

Point 8: Lines 84, 85: please use Arabic numerals instead of “twenty-one” and “two”.

Response 8: Thank you for your suggestion. We followed your suggestion.

Point 9: Line 88: spelling “The majority of newborn”

Response 9: Thank you for your suggestion. We corrected the spelling.

Point 10: Lines 88-89: the mean birthweight refers to all the study newborns, not only to those born at term. Please rephrase the sentence.

Response 10: Thank you for your suggestion. We rephrased the sentence as follows : Therefore, 280 infants met the eligibility criteria for this study, with a mean birth weight of 2744.6 ± 685.8 gram.

Point 11: Table 1: use footnotes for explanation of the data (n%?) and symbols.

Response 11: Thank you for your suggestion. We added footnotes.

Point 12: Lines 100, 109: “who” instead “that”

Response 12: Thank you for your suggestion. We replaced “that” by “who”

Point 13: Please use abbreviations when using a word for the first time in the text.

Response 13: Thank you for your suggestion. We checked our manuscript and explained the abbreviation where indicated.

Point 14: Line 138: level of bilirubin.

Response 14: Thank you for your remark, we made the adjustment.

Point 15: Line 151: blood taking? Better “blood collection

Response 15: Thank you, we changed blood taking to blood collection.

Reviewer 2 Report

This study investigates whether Kramer score on its own is sufficient in guiding the management of jaundice when it comes to phototherapy. The study has a nice sample size. A few suggestions in order to improve the manuscript:

  • It is still unclear what phototherapy threshold was in the study, or in clinical practice in general. If based on Kramer score, at what score do infant require phototherapy? If based on serum bilirubin level, at what bilirubin level do infants require phototherapy? The author needs to provide this information early on in the manuscript.
  • The author needs to discuss how phototherapy can reduce bilirubin level in blood. 
  • The author should discuss other ways to manage jaundice, such as IVIg and/or exchange transfusion.
  • The author should add graph legend to facilitate easier interpretation of the graph. The figure legend alone is not sufficient. 
  • I would suggest thorough grammar check throughout the manuscript. There are inconsistency with singular/plural, missing commas, etc. 

Author Response

Response to Reviewer 2 Comments

Point 1: It is still unclear what phototherapy threshold was in the study, or in clinical practice in general. If based on Kramer score, at what score do infant require phototherapy? If based on serum bilirubin level, at what bilirubin level do infants require phototherapy? The author needs to provide this information early on in the manuscript.

Response 1: We apologize we did explain clearly enough the use of the serum bilirubin and the use of the guideline in our paper. We changed the text in the method section to make it more clear. In short, Our study is a descriptive study. For each infant the Kramer score was determined. The serum bilirubin was measured at the same moment. We then plotted the result of the serum bilirubin on the graph of the most recent Indonesian guideline for the management of hyperbilirubinemia. This graph indicates if the infant needed phototherapy or not. For the analysis all infants were grouped according to their Kramer score. We hope that we clarified this issue, we added the above to the method section.

Point 2: The author needs to discuss how phototherapy can reduce bilirubin level in blood. 

Response 2: Many thanks for the advice. We have added in the discussion that phototherapy involves several biomolecular reactions, including photooxidation and photoisomerization. Phototherapy enables bilirubin to undergo photooxidation process in order to become quite small and polar to be excreted via urinary tract whilst photoisomerization reaction may take place faster by forming lumirubin, a constitutional isomer of bilirubin, which is more excretable in bile

Point 3: The author should discuss other ways to manage jaundice, such as IVIg and/or exchange transfusion.

Response 3: Many thanks for the advice. We added to the discussion the information about exchange transfusion and IVIGs as modalities other than phototherapy. 

Point 4: The author should add graph legend to facilitate easier interpretation of the graph. The figure legend alone is not sufficient. 

Response 4: Unfortunately it is not clear to us what the reviewer means with graph legend. The most important findings of the graphs is discussed in the result section, where we refer to the graph. Discussing the findings of the graph in the legend is not appropriate in our opinion. We welcome further instructions on this issue.

Point 5: I would suggest thorough grammar check throughout the manuscript. There are inconsistency with singular/plural, missing commas, etc.

Response 5: We checked the spelling and grammar also with a spelling check.

Round 2

Reviewer 1 Report

The revised version of the manuscript is very much improved; the most important issue is that the conclusion is supported enough by the results.